# Effect of sea surface temperature in El Niño regions on dengue dynamics in Colombia: Evidence from causal machine learning

Juan David Gutiérrez[ID]*, Johanna Tapias-Rivera[ID], Martha Liliana Hijuelos-Cárdenas, Ludivia Esther Montaño-Villalba[ID]

Universidad de Santander, Facultad de Ciencias Médicas y de la salud, Instituto Masira, Bucaramanga, Santander, Colombia

* jdgutierrez@mail.udes.edu.co

## Abstract

Dengue fever is among the most rapidly expanding vector-borne diseases globally, with Colombia ranking among the most affected countries in the Americas. Although previous research has linked climate variability and El Niño–Southern Oscillation (ENSO) episodes to dengue dynamics, the direct causal effect of sea surface temperature (SST) in El Niño regions remains insufficiently explored. We conducted a retrospective ecological analysis using monthly laboratory-confirmed dengue cases from 1,044 Colombian municipalities (2013–2023), combined with atmospheric, oceanic, and socioeconomic data. We emulated an experimental design to estimate the effect of SST in El Niño regions 1–2, 3, 3–4, and 4 on excess dengue cases. Confounder adjustment was guided by a Directed Acyclic Graph (DAG), and causal effects were estimated using Double Machine Learning (DML) with XGBoost learners. We estimated the Average Treatment Effect (ATE) and the Conditional Average Treatment Effect (CATE) conditioned on altitude. Robustness was evaluated with refutation tests introducing random confounders, subset replacement, and placebo exposures. A total of 455,329 confirmed dengue cases were reported during the study period, peaking in 2023. The strongest association was observed for El Niño region 3, where a standard deviation increase in SST (1.24 °C) raised the probability of excess dengue cases by 6.9 percentage points (ATE = 0.069, 95% CI = 0.056 – 0.083). El Niño regions 3–4 and 4 showed slightly weaker ATE yet significant effects (6.4% and 6.2%), while El Niño region 1–2 had the lowest effect (4.6%). The CATE analysis revealed that the effects of El Niño regions 1–2 and 3 were stronger at higher altitudes; meanwhile, for El Niño regions 3–4 and 4, the effects showed a slightly negative trend, suggesting a heterogeneous effect of El Niño regions on dengue incidence in Colombia based on altitude. Robustness checks indicated the presence of residual bias, particularly when applying the subset replacement test. These findings highlight the importance of integrating oceanic monitoring into early

**Data availability statement:** The code and dataset for replicating the results are accessible at: https://github.com/juandavidgutier/sst_dengue.

**Funding:** The authors received no specific funding for this work.

**Competing interests:** The authors have declared that no competing interests exist.

warning systems of the disease and tailoring vector-control strategies to local ecological contexts.

## 1. Introduction

Dengue fever is an acute viral illness transmitted by *Aedes aegypti* mosquitoes [1]. Among the diseases transmitted by mosquitoes, it is the one that spreads most rapidly worldwide [2]. According to the WHO, there are approximately 50–100 million reported cases annually, with 500,000 classified as severe, resulting in a mortality rate of approximately 2.5% [3]. According to WHO data, in Colombia by mid-2023, 50,818 cases of dengue had been reported, of which 25,958 (51%) were classified as severe dengue. These data exceed the cases reported for the same period in 2022 by 66% and the average of the last five years by 44% [4].

Exists substantial scientific evidence regarding the transmission dynamics of dengue in relation to environmental factors, particularly those related to temperature and precipitation [5–8], as well as climate variability and the El Niño phenomenon in Asia [9–11] and Latin America [12,13]. Sea surface temperature (SST) anomalies associated with the El Niño–Southern Oscillation (ENSO) are large-scale drivers of dengue dynamics, as they modulate atmospheric circulation and regional hydroclimatic conditions in continental regions, which, in turn, determine key parameters such as vector abundance [10,12].

Through teleconnections, SST anomalies in El Niño regions can produce spatially heterogeneous shifts in rainfall and raise ambient temperatures, both of which can accelerate *Aedes* population growth and shorten the extrinsic incubation period [14,15]. For surveillance and public-health practice, integrating SST/ENSO indicators into early-warning systems and predictive models can provide valuable lead time for targeted vector control and resource allocation [13], but operationalization requires coupling oceanic predictors with context-specific data to account for effect heterogeneity and avoid false alarms.

Causal machine learning represents a paradigm shift in epidemiological research, combining the predictive power of machine learning algorithms with the theoretical framework of causal inference to address complex causal relationships between environmental exposures and disease outcomes [16]. Traditional parametric statistical methods rely on correct model specification assumptions—which are particularly challenging in environmental epidemiology settings.

In the context of estimating the causal effect of SST in El Niño regions on dengue incidence, causal machine learning methods provide superior flexibility to capture complex non-linear relationships, interactions between climatic variables, and high-dimensional confounding structures without imposing restrictive functional form assumptions that traditional regression approaches require [17,18].

This study aims to estimate the effect of SST in El Niño regions on the incidence of dengue cases in Colombia between 2013 and 2023. Implementing causal inference and machine learning methods, we aim to disentangle the interplay between oceanic processes in El Niño regions and socio-environmental disease dynamics, providing

evidence-based insights for targeted public health interventions and public health management strategies to mitigate dengue transmission in Colombia and similar endemic regions.

## 2. Methods

### 2.1. Ethics statement

Given the anonymized nature of the data, the Institutional Review Board of the Universidad de Santander approved the research (Minutes of the Bioethics Committee VII – 005 – BUC), as stipulated in the Common Rule (45 CFR §46). The study conforms to the Strengthening the Reporting of Observational Studies in Epidemiology reporting guidelines. The data on dengue cases were sourced from anonymized information; therefore, the data available in the GitHub repository do not include any details that could identify participants. The dataset in the repository is composed solely of aggregated, non-identifiable information. All personal identifying details were removed during preprocessing to adhere to data protection laws and ethical research guidelines.

### 2.2. Data collection

**2.2.1. Data cases.** Dengue case data were supplied (December 3, 2024) by the National Public Health Surveillance System of Colombia (SIVIGILA), encompassing laboratory-confirmed cases (i.e., IgM ELISA and RT-PCR tests) between January 2013 and December 2023. Data were aggregated at the municipality level monthly. Records with inconsistencies in reporting dates, municipality occurrence codes, or age (e.g., > 120 years) were excluded. To minimize the inclusion of potentially allochthonous cases, an altitudinal exclusion criterion was applied: cases reported from municipalities located above 2,200 meters above sea level (masl) were removed, following the established altitudinal threshold for dengue transmission established by the National Institute of Health, which corresponds to the biological limits of *Aedes aegypti* in the country [19].

Monthly expected dengue cases and the Standardized Incidence Ratio (SIR), per municipality, were estimated using the epitools package (version 0.5-10.1) [20]. Incidence rates were age-standardized using the indirect standardization method [21], based on the WHO's age group classifications [22] and population projections from the Colombian National Statistics Department [23]. To estimate excess dengue cases, we binarized the SIR, assigning a value of 1 when the SIR was greater than 1 (indicating excess cases) and 0 otherwise.

**2.2.2. Environmental data.** We retrieved 14 atmospheric and oceanic monthly indices (Table 1) from the National Oceanic and Atmospheric Administration (NOAA) database, covering the period from January 2013 to December 2023 [24]. Note that in Table 1, the indices SST12, SST3, SST34, and SST4, correspond to the El Niño regions.

Each climate index in Table 1 represents a distinct large-scale ocean–atmosphere process that influences regional hydroclimate and, consequently, dengue transmission. The indices capture Pacific SST variability, which modulates hydro-climate variables through well-established teleconnections [25]. Meanwhile, Atlantic SST indices have been suggested to be associated with ENSO [26,27], along with changes in hydro-climate patterns, particularly in the northern region of the country. Together, these indices account for major, partially independent sources of SST in El Niño regions that may simultaneously influence both local environmental conditions and dengue outcomes over seasonal to interannual timescales.

Monthly data spanning from 2013 to 2023 on rainfall and temperature were downloaded from the ERA5 dataset of the European Space Agency, which offers a spatial resolution of 0.10 degrees. To perform spatial matching of rainfall and temperature variables, we utilized the extract function from the raster package in R (version 4.0.3) [26]. This procedure identified the pixels within each municipal polygon and subsequently computed the average values of each variable for each municipality and month.

Data of the minimum altitude of each municipality was obtained from the National Geographic Institute Agustín Codazzi [27].

Global Public Health

**Table 1. Atmospheric and oceanic indices included in the study.**

| Name | Detail |
|---|---|
| SST12 | Sea surface temperature in El Niño region 1–2 |
| SST3 | Sea surface temperature in El Niño region 3 |
| SST34 | Sea surface temperature in El Niño region 3–4 |
| SST4 | Sea surface temperature in El Niño region 4 |
| NATL | Sea surface temperature North Atlantic (5–20°North, 60–30°West) |
| SATL | Sea surface temperature South Atlantic (0–20°South, 30°West-10°East) |
| TROP | Sea surface temperature Global Tropics (10°South-10°North, 0–360) |
| SOI | (Standardized anomalies Tahiti - Standardized anomalies Darwin) sea level pressure |
| ESOI | Indonesia sea level pressure (Standardized Anomalies) |
| CPOLR | Central Pacific outgoing long wave radiation (170°E-140°W,5°S-5°N) |
| WPAC850 | West Pacific 850 mb trade wind index (135°East-180°West, 5°North-5°South) |
| CPAC850 | Central Pacific 850 mb trade wind index (175°West-140°West, 5°North-5°South) |
| EPAC850 | East Pacific 850 mb trade wind index (135°West-120°West, 5°North-5°South) |
| QBO_u30 | 30 mb Singapore winds |

mb = millibar.

We implemented the definitions of El Niño and La Niña episodes developed by the NOAA [24]. According to NOAA, a month is classified as part of an El Niño or La Niña event based on the three-month moving average of SST in the El Niño 3–4 region. For El Niño, this average must be at least +0.5 °C or higher, while for La Niña, it must be -0.5 °C or lower. S1 Table details the classification of each month from 2013 to 2023 as part of an El Niño, La Niña, or Neutral episode according to the NOAA.

**2.2.3. Socio-economic data.** The percentage of households in each municipality with multidimensional poverty index (MPI) was sourced from the Terridata repository [28]. The MPI is a national socioeconomic index collected during the last national census of 2018. It encompasses six dimensions: education access, childhood and youth conditions, work opportunities, health access, public services, and housing quality. To estimate the population density for each municipality per year, we divided the population of each municipality by the corresponding polygon's area. Yearly population data were based on national population estimates obtained from the National Statistics Department [23].

## 2.3. Definition of variables in the study

The outcome variable in our study was the excess of dengue cases in each municipality and each month between 2013 and 2023, which is a binary variable. We used the SST in El Niño regions as the treatment variable. Note that we emulated (see 2.4 Causal inference analysis) four experimental designs, each evaluating the effect of SST in each El Niño region on excess dengue cases.

## 2.4. Causal inference analysis

First, note that causal inference aims to emulate an experimental design, using an observational dataset, to estimate the effect of the exposure variable on the outcome to avoid inducing other forms of bias (e.g., collider bias) and identifying the correct adjustment for effect estimation, based on the relationships between the variables included in the model. We employed a Directed Acyclic Graph (DAG) to visually represent our prior knowledge about the relationships between the variables and to explicitly state our causal assumptions [29]. In our DAG, for all emulated experimental designs, we included the SST in an El Niño region as the exposure variable and excess dengue cases as the outcome (Fig 1). Year

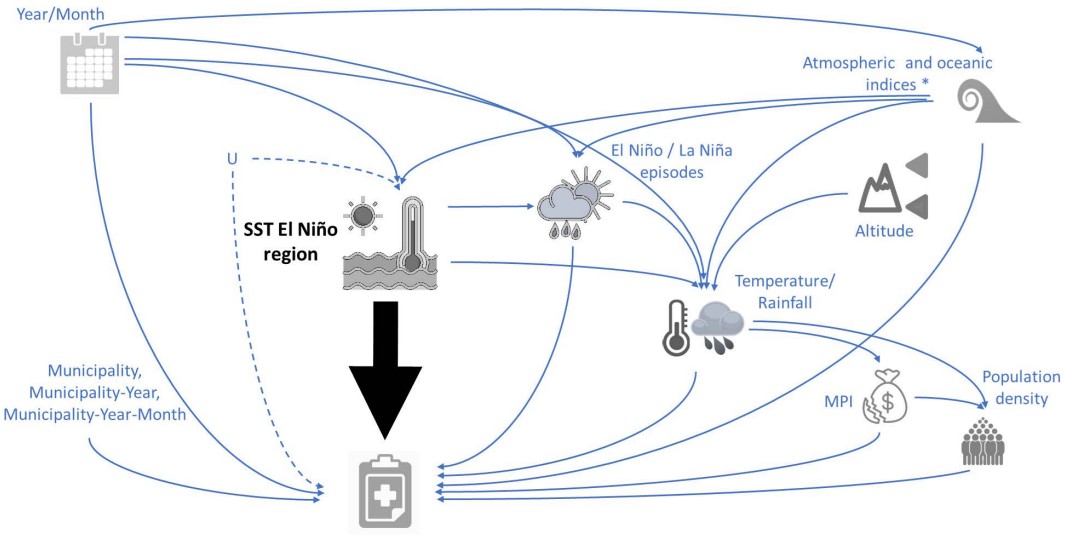

**Fig 1. Directed Acyclic Graph (DAG) depicting the hypothesized causal relationships between sea surface temperature (SST) in El Niño regions (exposure variable) and excess dengue cases (outcome variable).** The black arrow signifies the causal association of interest. The node labeled U represents potential unobserved confounders, which could introduce remaining confounding bias into our estimate of the effect of SST in El Niño regions on excess dengue cases. MPI represents multidimensional poverty index. The asterisk on the node atmosperic and oceanic indices represents the association between the variables included in this node. Municipality=Descriptor variable for each municipality. Municipality-Year=Descriptor variable for each municipality in each year. Municipality-Year-Month=Descriptor variable for each municipality in each year and each month. The figure was created using open-source images from Openclipart (https://openclipart.org), which are released into the public domain under the CC0 1.0 license (https://openclipart.org/faq).

and month were included as potential confounders because long-term and seasonal trends in the SST and case numbers can be driven by these variables.

Note that hydro-climate patterns in continental regions are driven by oceanic and atmospheric conditions [30–32]. In that sense, atmospheric and oceanic indices were also considered as potential confounders since they simultaneously influence the monthly SST patterns in El Niño regions [33–35] and can affect dengue incidence through other climate variables, such as relative humidity [14]. Rainfall and temperature were included as potential effect modifiers because the effect of SST in El Niño regions on dengue incidence occurs through these variables. Note also that both variables (rainfall and temperature) are also associated with dengue incidence [36]. Additionally, the MPI and population density were included in our analysis as potential effect modifiers. Note that altitude is a proxy for average temperature, and that high altitude corresponds to lower average temperatures in the municipalities.

The node of the atmospheric and oceanic indices in our DAG has an asterisk, representing the global teleconnections and interdependencies among the indices [25]. In our DAG, the nodes Rainfall and Temperature are associated with the nodes MPI and population density, because historically, since the arrival of Spanish colonizers to Colombia, municipalities with extreme climates (e.g.,: high temperature and heavy rainfall) have been recognized as regions with unhealthy climates, leading these areas to become settlements mainly of ethnic minorities such as indigenous and Afro-descendant communities. Consequently, these territories have been systematically neglected by the government established in Bogotá, the capital city. Perhaps the most notable examples are the regions of the Amazon and Pacific, characterized by high rainfall, and a very low presence of the Colombian state in terms of social development and provision of necessities in these communities [37].

To facilitate the convergence of the machine learning models (see 2.5 Machine learning implementation), we estimated the Average Treatment Effect (ATE) for each increase of one standard deviation (sd) unit of temperature for the four

emulated experimental designs. We estimated the effect of exposure to the average SST in El Niño regions for the current and previous month, covering the time necessary for changes in the vector's biology and population, as well as the extrinsic incubation period, intrinsic incubation period, and the reporting of new cases. The ATE represents the average change in the probability of having an excess of dengue cases for each increase of one sd in SST in El Niño regions. Subsequently, we estimated the Conditional Average Treatment Effect (CATE) conditioned on altitude. Note that the CATE is the ATE given the heterogeneity in altitude (also, in terms of average temperature, altitude serves as a proxy for the average temperature in the municipalities), i.e., the ATE of SST in El Niño regions on excess dengue cases among the municipalities with the same altitude value. Note that when we estimated the effect of SST in El Niño region 1–2 on excess dengue cases, we included the other SSTs in El Niño regions (i.e., SST3, SST3–4, and SST4) as potential confounders. We followed the same logic when estimating the effects of SST3, SST3–4, and SST4.

Our estimates of the effect of SST in El Niño regions on excess dengue cases would be interpreted as causal (i.e., free of bias) under the following assumptions [38,39]: 1) Unconfoundedness: Given all potential confounders, the outcome is independent of the exposure. 2) Consistency: The exposure variable is sufficiently well-defined such that any variation within the definition of the exposure would not result in a different outcome, and that there is no interference. 3) Positivity: There exists a probability greater than 0 of observing each value of exposure within each combination of covariates among the studied units. 4) No measurement error: No substantial measurement errors are present such that no substantial measurement bias is induced. 5) Well specified model: The model is well-specified, such that all relevant non-linearities and/or statistical interactions are taken into account.

We employed the R package DAGitty (version 0.3-1) [40] to assess the consistency between the assumptions of the DAG and our dataset (i.e., the conditional independences). The appropriate adjustment to the causal model was identified using the DoWhy Python package (version 0.11.1) [41].

## 2.5. Machine learning implementation

We implemented a sparse version of the Double Machine Learning (DML) algorithm to estimate both the ATE and CATE of SST in El Niño regions on excess dengue cases. The DML framework addresses confounding bias by using machine learning models to estimate both the outcome and treatment assignment mechanisms simultaneously, thereby providing causal effect estimates in the presence of high-dimensional confounders. Our implementation employed 5,000 trees in a XGBoost framework with a learning rate of 0.0001, as the base learners for both the outcome model and treatment model, which were fitted using 5-fold cross-validation to prevent overfitting, and a maximum of 30,000 iterations to ensure convergence. Altitude was maintained in its original continuous scale to serve as the primary effect modifier for heterogeneous treatment effect estimation, allowing us to examine how the effect of SST in El Niño regions on dengue varies across different altitude conditions (i.e., the CATE).

To estimate the CATE and after fitting the DML model, we generated a grid of 100 evenly spaced altitude values spanning the observed range within the data. To estimate the CATE at each point along this altitude grid, we held the other effect modifier variables constant at their respective mean values observed in the dataset. Given that the fixed effect variables (i.e., municipality, municipality-year, and municipality-year-month) required normalization for inclusion in the model due to computational requirements in Python, these were held at their mean values during the estimation. This methodological approach allows us to isolate the marginal effect of altitude, illustrating how the effect of SST in the El Niño regions on excess dengue cases varies with altitude while controlling for the spatio-temporal structure of the data. For each altitude value in the grid, we predicted the pointwise treatment effect, resulting in a continuous CATE curve. Simultaneously, we calculated the corresponding 95% confidence interval (CI).

Month values (1–12) were addressed through cyclic encoding of monthly seasonality using sine and cosine transformations, which preserved the cyclical nature of seasonal variation without introducing artificial ordinality. After identifying the appropriate adjustment to estimate the effect, avoiding inducing bias, all confounding variables, except year and month

indicators, were binarized using median splits to reduce the influence of outliers and create more stable treatment effect estimates across different subgroups of the population.

To address the remaining confounding bias, we incorporated SST values for the subsequent month as a negative control exposure. Because of temporal ordering, future SST values cannot causally influence dengue cases that have already occurred. We followed the methodology proposed by Flanders et al. [42], which utilizes future exposure that does not influence prior effects, minimizing unobserved confounding bias [43].

## 2.6. Robustness tests

It is crucial to recognize that an exemplary epidemiological study is one devoid of bias, meaning the estimated effect can be deemed genuinely causal [29,38,39]. Nonetheless, all observational epidemiological studies are prone to various biases, such as misclassification bias, confounding bias, selection bias, information bias, collider bias, and inference bias, among others [43]. In this research, we have utilized several approaches, including causal machine learning techniques, to address these biases. It is important to note that if our methods for addressing these biases were entirely effective, all forms of bias would be removed, and no bias signals would appear in robustness tests. However, our estimates might still be affected by some of these biases. To this goal, we performed a series of robustness tests to identify any remaining bias in our estimates, even after employing multiple strategies to mitigate different bias sources.

We performed a series of robustness tests to assess how resilient our causal estimates are to potential biases. These tests are intended to examine the credibility of the estimated causal effect by intentionally introducing variations in the estimation process. The purpose is to determine if the estimated effect undergoes significant changes in situations where no actual causal effect should be present or where the estimate should remain consistent. In this scenario, remaining bias refers to any systematic deviation in the estimated effect caused by various sources of bias that could undermine the causal interpretation. A robust causal model should provide consistent estimates despite perturbations and yield null effects when tested in simulated null scenarios.

In our analysis, we set the argument num_simulations = 50 for all robustness tests. This argument defines how many times the refutation procedure is conducted to evaluate the robustness of the estimated causal effect. Throughout these tests, we employed the Bootstrap method to estimate the frequency with which the test value appears in the null distribution. This null distribution represents the same quantity generated under specific conditions that differ from one test to another, as recommended by Rousselet et al. [44]. The resulting frequency is interpreted as a probability, yielding a p-value for the null hypothesis. For each test, the p-value determines whether the new effect estimate, after perturbation, significantly differs from the original ATE estimate (or from zero in the case of the placebo test). A p-value < 0.05 suggests that there is still bias present in the original estimate.

We evaluated the estimates through three robustness tests, useful for causal machine learning contexts: 1) Adding a random common cause: This simulates the introduction of an unrelated factor that might influence both the treatment and outcome variables. A valid estimate should remain largely unaffected. 2) Replace a small portion (10%) of the data with random values. A robust effect estimate should be minimally affected by such data modification. 3) Adding a placebo treatment: This introduces a fictitious treatment group with no actual intervention. A valid estimate should show no difference between the treatment and placebo groups.

Note that adding a random common cause and the replacement of a portion of the data corresponds to a synthetic negative control with an invariant transformation, whereas adding a placebo treatment corresponds to a synthetic negative control with a nullifying transformation [44].

The ATE and CATE estimation were conducted using the package EconML (version 0.15.1) [45] in Python. Refutation tests were developed with the DoWhy Python package. The code and dataset for replicating the results are accessible at: https://github.com/juandavidgutier/sst_dengue.

## 3. Results

Between January 2013 and December 2023, there were 455,329 laboratory-confirmed cases of dengue reported in the 1,044 municipalities included in the study, with the majority of cases occurring in males (56.6%, n = 257,559). As we expected, the three largest cities in our study reported the largest number of cases: Cali (40,902), Barranquilla (19,674), and Medellín (15,600). The years with the most cases were 2023 (83,708), 2019 (60,655), and 2013 (56,007) (Fig 2a). All municipalities where the altitude makes dengue transmission unsuitable are located in the Andean region, mainly in the central part of the country (Fig 2b).

During the study period, the minimum SST in the El Niño regions was 19.1 °C, and the maximum SST was 30.2 °C. The region with the lowest temperature values was El Niño region 1–2 (Fig 3a), while the region with the highest temperature was El Niño 4 (Fig 3d). The El Niño regions 3 and 3–4 showed temperatures between 24.1 and 29.5 °C (Fig 3b and 3c).

The analysis of the DAG allowed us to identify that the variables: MPI, population density, rainfall, temperature, and El Niño and La Niña episodes, are colliders (i.e., variables influenced by two or more covariates), and these variables were excluded from the estimation of the effect of the SST in El Niño regions on excess dengue cases, to avoid inducing collider bias. In our causal framework, these collider variables are jointly influenced by several upstream drivers. Conditioning on such variables would open non-causal paths between the exposure and the outcome, inducing spurious associations even in the absence of a true effect. To avoid this collider bias, we therefore exclude these downstream variables from adjustment and focus only on valid confounders that precede both exposure and outcome [29,46].

The largest ATE was observed for the El Niño region 3, where a rise of one sd (1.24 °C) increases the probability of excess dengue cases by 6.9 percentage points (ATE = 0.069, 95% CI = 0.056 – 0.083). For the El Niño regions 3–4 and 4, the ATE was 0.064 (95% CI = 0.051 – 0.078) and 0.062 (95% CI = 0.048 – 0.075), respectively, when the SST increases by one sd (i.e., 0.97 °C and 0.68 °C, respectively). The lowest observed ATE corresponded to El Niño region 1–2, where an increase of one sd (2.33 °C) corresponds to an increase in the probability of excess dengue cases of 4.6 percentage points (ATE = 0.046, 95% CI = 0.032 – 0.059) (Fig 4). Note that El Niño region 1–2 is the closest and that El Niño region 4 is the farthest from Colombia.

It is important to understand that the ATE values represent shifts in probability at the municipal level. For instance, an ATE of 0.069 for the effect of El Niño 3 region specifies that a 1.24 °C increase in SST (i.e., one sd) boosts the likelihood of a municipality experiencing excess cases (SIR > 1) by 6.9 percentage points. Considering all 1,122 municipalities in Colombia, this equates to approximately 77 additional municipalities (1,122 × 0.069 ≈ 77 municipalities below 2,200 meters above sea level), moving to excess cases for each sd rise in SST in El Niño region 3.

The CATE conditioned on altitude for the SST in El Niño regions showed a trend toward an increase in regions 1–2 and 3, with a more evident pattern in region 1–2 (Fig 5). In both cases, a higher altitude (corresponding to lower average temperature in the municipalities) was associated with a stronger effect of El Niño regions 1–2 and 3 on excess dengue cases (Fig 5a and 5b). By contrast, for El Niño regions 3–4 and 4, the CATE conditioned on altitude showed a minor trend toward a negative association, where higher altitude was linked to a slightly weaker effect of these regions on excess dengue cases (Fig 5c and 5d).

When we applied the robustness tests by adding a random common cause and a placebo treatment for all El Niño regions, these tests produced p-values > 0.05. However, the p-values for all El Niño regions in the robustness test of replacing a random subset were always < 0.05. These last results suggest the presence of remaining bias in our estimates of the effect of SST in El Niño regions on excess dengue cases in Colombian municipalities Table 2.

The GitHub repository https://github.com/juandavidgutier/sst_dengue includes the code and dataset to replicate the results.

## 4. Discussion

Our results suggest a significant ATE of SST in all El Niño regions on excess dengue cases in Colombian municipalities suitable for the transmission of the disease. However, the CATE based on altitude revealed a different

a

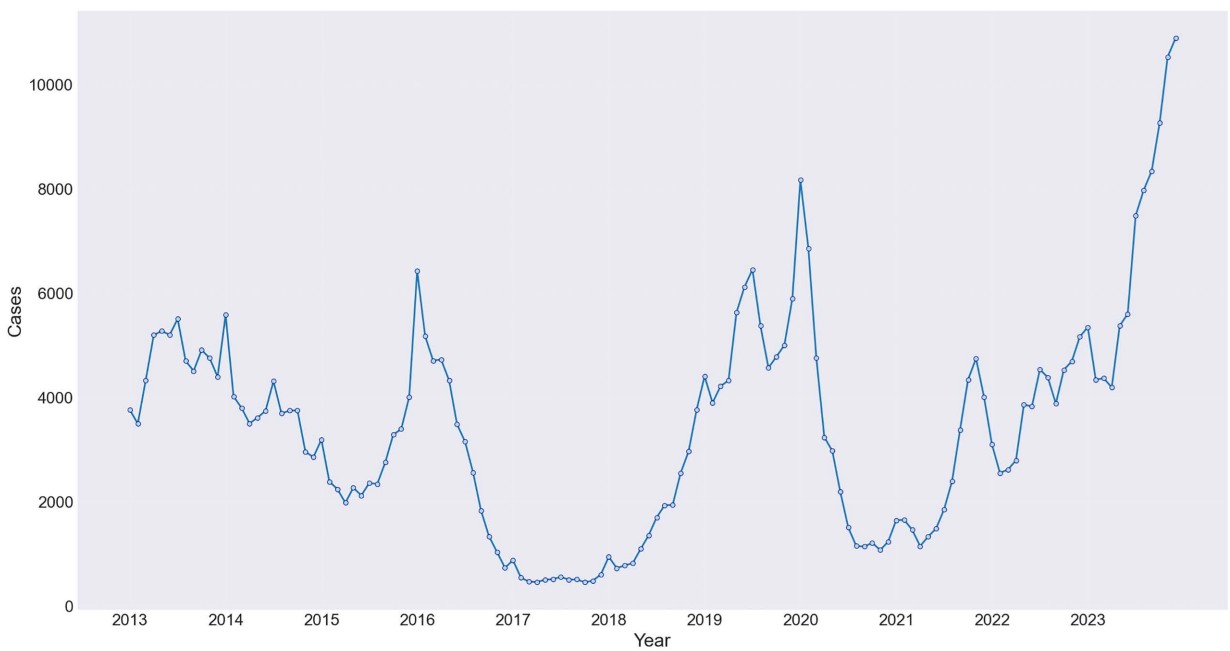

b

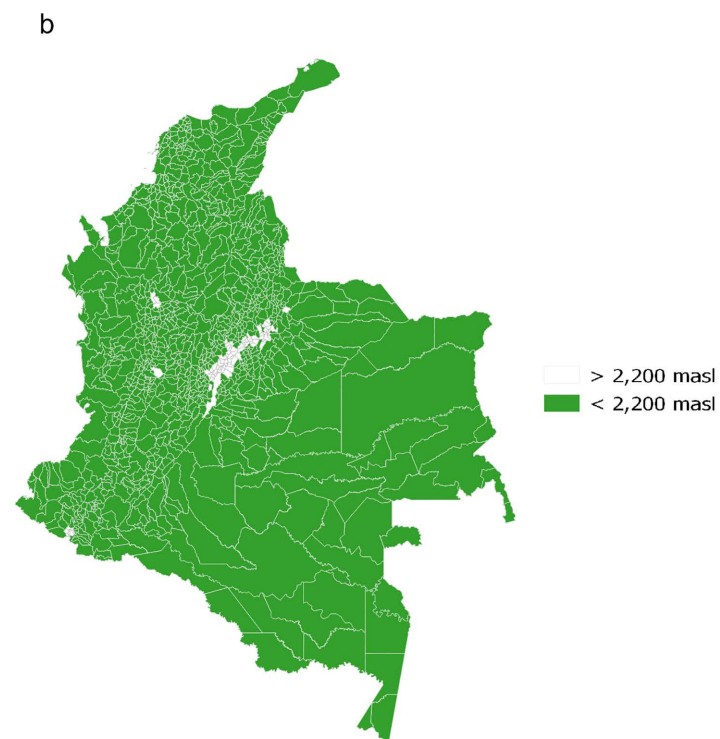

**Fig 2. Monthly time series of dengue cases confirmed by laboratory tests in Colombia between 2013 and 2023 (a), and municipalities with altitudinal suitability for dengue transmission (i.e.,<2,200 meters above sea level (masl)) (b).**The map was created using QGIS software, with the

basemap shapefile sourced from the Colombian National Geostatistical Framework, an openly available resource (https://www.dane.gov.co/files/geoportal-provisional). The terms of use for the base shapefile are compatible with CC-BY 4.0 (https://geoportal.dane.gov.co/acerca-del-geoportal/licencia-y-condiciones-de-uso/#gsc.tab=0).

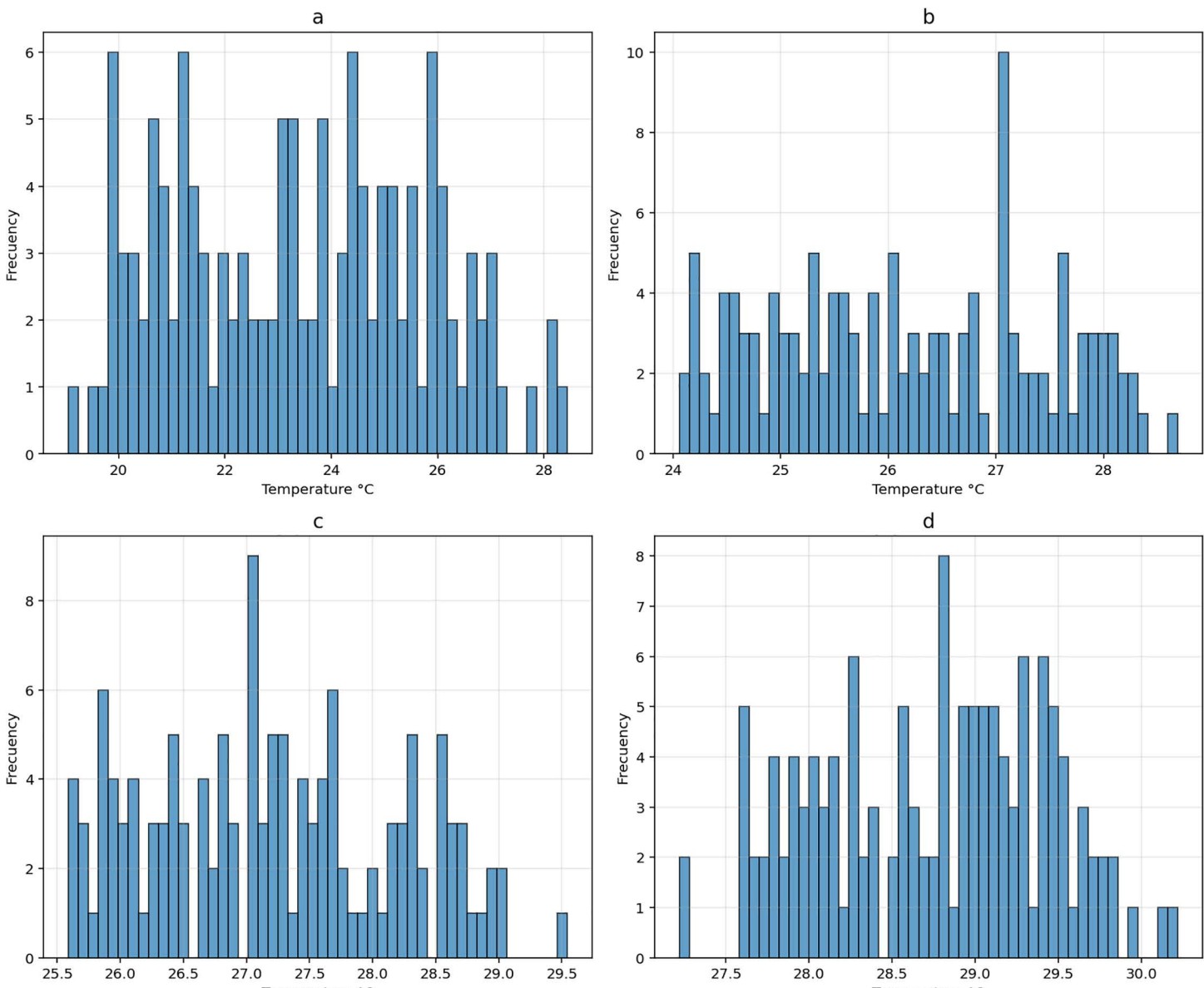

**Fig 3. Histograms of the sea surface temperature (SST) between 2013 and 2023 for El Niño regions: a) El Niño region 1–2; b) El Niño region 3; c) El Niño region 3–4; d) El Niño region 4.**

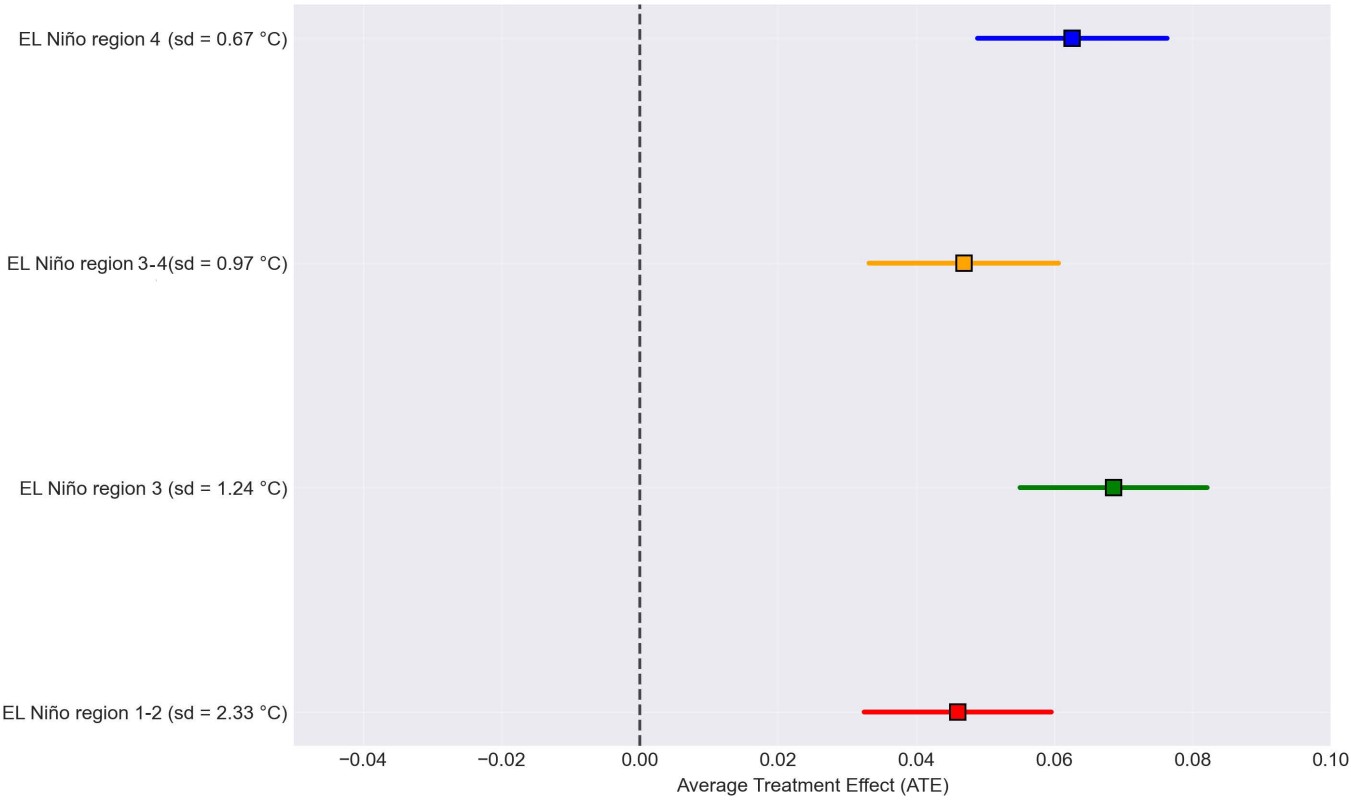

**Fig 4. Average Treatment Effect (ATE) of the SST in El Niño regions on excess dengue cases between 2013 and 2023.** The squares represent the point estimates, and the horizontal lines indicate the 95% CI. The vertical black line represents the null effect, where the ATE = 0.

pattern, where higher altitudes were associated with a stronger effect from El Niño regions 1–2 and 3 on excess dengue cases, and a trend toward a slightly negative or no effect from El Niño regions 3–4 and 4 on excess dengue cases in higher altitudes, a pattern never before documented. The underlying mechanism for this different pattern of CATEs remains unclear, and it is likely that the slight negative association observed actually corresponds to a non-association between the effect of SST in El Niño regions 3–4 and 4, and the altitude of the municipalities, because these regions are the farthest from Colombia. Similarly, we hypothesize that at higher altitudes (lower temperatures), thermal conditions tend to be suboptimal for the vector [47], but the rise of SST in El Niño regions 1–2 and 3 could create microclimates in the municipalities that facilitate breeding sites for the vector and the transmission of dengue.

The results obtained in our study support the evidence of a positive association between increased SST in El Niño regions and dengue incidence in Latin America, although the magnitude and response time vary according to geographic and temporal contexts [15,36,48–51]. Most previous studies include predictive models and time series, and suggest that SST can anticipate dengue outbreaks several weeks or months in advance [50,52–54]. However, the relationship is not uniform; local factors such as urbanization, immunity, and interventions can modulate or even attenuate this association [55–58]. Moreover, some studies report negative or null associations in certain regions or years, underscoring the importance of considering the local context and the intensity of the El Niño episodes [48,51,59].

The reliance on secondary data from SIVIGILA, a passive surveillance system, introduces potential information bias into our estimation of the causal effect. Passive surveillance depends on healthcare providers reporting

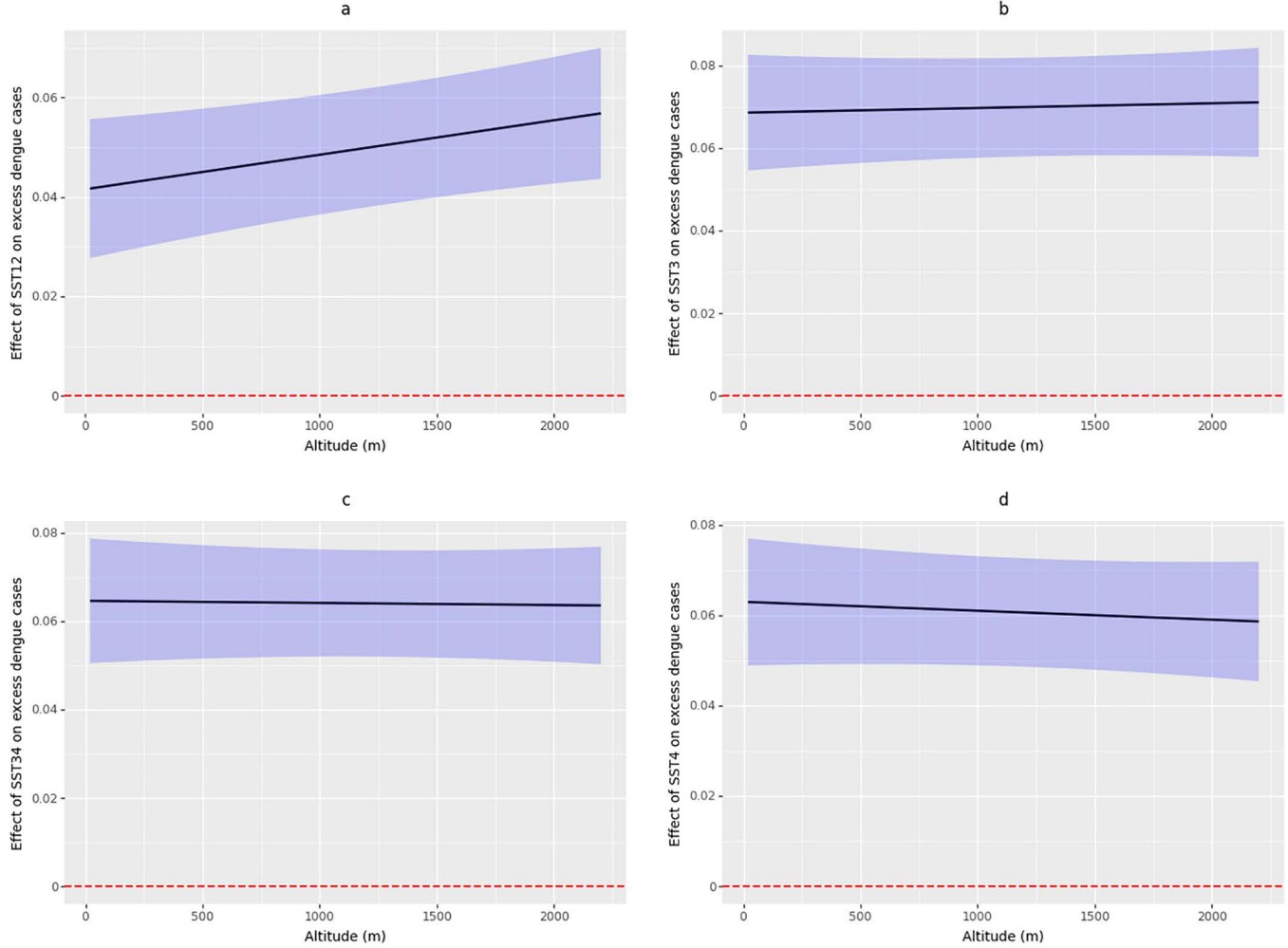

**Fig 5. Conditional Average Treatment Effect (CATE) estimates, conditioned on altitude (X-axis), for the SST in El Niño regions: a) El Niño region 1–2; b) El Niño region 3; c) El Niño region 3–4; and d) El Niño region 4.** The black line represents the average effect, the blue shaded area represents the 95% CI, and the dotted red line indicates the null effect (CATE = 0).

laboratory-confirmed cases, which can lead to inconsistent reporting patterns. In years with dengue outbreaks, heightened awareness and increased diagnostic testing often result in more cases being laboratory-confirmed and reported.

In contrast, during non-outbreak periods, many cases may not undergo laboratory testing and thus remain unreported in SIVIGILA, underestimating the true disease burden. This variability in reporting practices can bias our estimates of the ATE and CATE. Studies have underscored the limitations of passive surveillance systems like SIVIGILA, noting that they can miss significant numbers of cases and that reporting is influenced by healthcare-seeking behavior and diagnostic capacity [60,61]. However, there is no data on the variability in laboratory confirmation rates between years or underreporting during healthcare disruptions.

The heterogeneity of municipalities in Colombia, particularly those altitudinally suitable for dengue transmission, poses challenges for epidemiological surveillance and may contribute to less consistent data across our study. Municipalities differ substantially in healthcare infrastructure and access to diagnostic resources, all of which can affect the quality and reliability of SIVIGILA data. For instance, small municipalities with limited healthcare facilities may underreport cases, while

**Table 2. Refutation tests assessing the effect of SST in El Niño regions on excess dengue cases.** Note that we ran each robustness test 50 times and obtained a distribution of results after perturbation of the corresponding refutation test (New effect values in the Table). This distribution was then compared with the distribution of the ATE obtained in the original estimation (Estimated effect values in the Table), using a null hypothesis of no difference. A p-value < 0.05 indicates statistical evidence of a difference relative to the original ATE estimation and suggests the presence of remaining bias. Statistical significance is indicated in parentheses (p-value); p-values < 0.05 are shown in bold. Note that the units in the table are the same as those of the ATE, i.e., change in the probability of excess dengue cases.

| Exposure | Add a random common cause (p-value) | Replace a random subset (p-value) | Add a placebo treatment (p-value) |
|---|---|---|---|
| SST in El Niño region 1–2 | Estimated effect = 0.046 | Estimated effect = 0.046 | Estimated effect = 0.046 |
|  | New effect = 0.046 (0.06) | New effect = 0.065 (**< 0.01**) | New effect = 0.000 (0.44) |
| SST in El Niño region 3 | Estimated effect = 0.069 | Estimated effect = 0.069 | Estimated effect = 0.069 |
|  | New effect = 0.069 (0.52) | New effect = 0.081 (**< 0.01**) | New effect = 0.000 (0.47) |
| SST in El Niño region 3–4 | Estimated effect = 0.064 | Estimated effect = 0.064 | Estimated effect = 0.064 |
|  | New effect = 0.064 (0.34) | New effect = 0.057 (**< 0.01**) | New effect = 0.000 (0.49) |
| SST in El Niño region 4 | Estimated effect = 0.062 | Estimated effect = 0.062 | Estimated effect = 0.062 |
|  | New effect = 0.062 (0.44) | New effect = 0.041 (**< 0.01**) | New effect = 0.001 (0.39) |

SST = Sea surface temperature.

those with better resources may capture data more accurately. This inconsistency can introduce bias into our machine learning models, which rely on sufficient and representative samples for training and validation, potentially skewing the estimated effects of SST in El Niño regions on dengue incidence. Previous research has highlighted how such heterogeneity in Colombian municipalities leads to variable surveillance outcomes, attributing these differences to disparities in healthcare access and reporting practices [62].

The findings from the refutation test of replacing a subset indicate that our results should be interpreted with caution due to the possible presence of effect-modifying factors that were not accounted for in the analysis. One such factor is seasonal variations in healthcare accessibility, which could influence dengue case reporting. For example, during dry season and periods of low rainfall, healthcare facilities may be more accessible, leading to increased reporting of cases; conversely, during high rainfall periods, flooding might reduce access to healthcare services, resulting in fewer reported cases.

A previous study by Delmelle et al. [63] in Cali, Colombia, underscored how healthcare utilization patterns vary and could impact case reporting. Carabali et al. [64] demonstrated significant underreporting of dengue in Colombia's national surveillance system, particularly during periods of healthcare strain, suggesting that fewer reported cases during extreme rainfall might reflect surveillance disruptions rather than reduced incidence. Although our study adjusted for potential confounders, it did not account for these latent and fine-scale factors, and future research should integrate data on healthcare access and reporting biases to refine causal effect estimates.

The failure of the refutation test of replacing a subset suggests that the causal effect estimate may be sensitive to unmeasured behavioral patterns, such as climatic conditions and local water storage practices, that might influence the relationship between SST in El Niño regions and dengue transmission. For instance, studies have shown that low rainfall conditions can increase dengue risk by promoting stagnant water accumulation in containers, a key breeding site for *Aedes aegypti* mosquitoes [65].

The municipal scale used in our study presents limitations in capturing the fine-scale heterogeneity inherent in dengue transmission dynamics. Dengue transmission is highly localized, primarily due to the limited flight range of its vector, *Aedes aegypti*, which typically disperses less than 200 meters [66]. Previous studies have shown that dengue incidence often clusters at the neighborhood or household level, a granularity that our municipality resolution cannot adequately resolve [67,68].

Finally, the data in this study were grouped by municipality, potentially introducing within-group correlations that violate the independent and identically distributed assumption underlying the causal machine learning methods employed, such as those implemented in the package EconML [45]. This assumption may not hold due to shared characteristics or conditions within municipalities, which could lead to underestimated standard errors and overconfident inferences, as well as contribute to the observed overfitting indicated by the failed refutation test.

Mixed-effects models, which incorporate random effects to account for such hierarchical structures, are widely used in dengue research to model spatiotemporal patterns and environmental influences [8,69]. However, current causal machine learning frameworks do not integrate mixed-effects structures, limiting their applicability to clustered data and potentially affecting the reliability of the estimated causal effects. Future research could explore combining mixed-effects models with causal machine learning to improve robustness when analyzing grouped data [70].

## 5. Conclusion

This study uncovers a complex relationship between SST in El Niño regions and dengue transmission in Colombia over the period from 2013 to 2023. Through the application of causal machine learning techniques, we identified an effect of SST in all El Niño regions on excess dengue cases. These insights emphasize the need to account for oceanic monitoring in the development of effective dengue surveillance and risk prediction systems, especially in regions like Colombia, where such environmental dynamics can be critical.

The findings of this research carry significant implications for public health and environmental monitoring strategies. By pinpointing the magnitude of the effect of SST in El Niño regions associated with heightened dengue risk, our study suggests that targeted interventions—such as enhanced vector control measures during periods of increase in SST in El Niño regions—could prove effective in curbing transmission. Additionally, to the monitoring of the SST in El Niño regions, optimizing urban sanitation and reducing social vulnerability of communities could serve as a preventive measure to reduce dengue incidence. These observations call for a more integrated approach, where environmental monitoring practices are seamlessly coordinated with public health efforts. This framework could extend beyond Colombia, offering a replicable strategy for dengue monitoring in other regions with comparable climatic profiles, where vector-borne diseases remain a pressing concern.

While our methodological approach is robust, certain limitations should be acknowledged. Firstly, our estimate of the effect of SST in El Niño regions on excess dengue cases cannot be considered causal and free of bias. The possibility of remaining bias, as hinted by robustness analyses, indicates that unmeasured factors, such as access to healthcare, might have influenced our conclusions. Moreover, the broad spatial resolution of the data may have masked localized variations in dengue transmission patterns, which are often driven by the restricted movement of the *Aedes aegypti* mosquito. To address these gaps, future studies should leverage finer-scale data and explore ways to account for these hidden effect modifiers, refining the accuracy of causal relationships. Integrating real-time oceanic monitoring models with disease surveillance systems could further enhance predictive capabilities, shedding light on the intricate interplay between SST in El Niño regions, climate, and dengue transmission. This research thus lays a foundation for deeper investigations into the oceanic drivers of vector-borne diseases, underscoring the value of multidisciplinary strategies in adapting to a warming world.

## Supporting information

**S1 Table. The table shows the episodes according to the National Oceanic and Atmospheric Administration (NOAA).**
(XLSX)

## Acknowledgments

We thank the Colombian Ministry of Health for providing access to epidemiological data.

## Author contributions

**Conceptualization:** Juan David Gutiérrez, Johanna Tapias-Rivera, Martha Liliana Hijuelos-Cárdenas, Ludivia Esther Montaño-Villalba.

**Data curation:** Juan David Gutiérrez, Johanna Tapias-Rivera, Martha Liliana Hijuelos-Cárdenas, Ludivia Esther Montaño-Villalba.

**Formal analysis:** Juan David Gutiérrez, Johanna Tapias-Rivera, Martha Liliana Hijuelos-Cárdenas, Ludivia Esther Montaño-Villalba.

**Funding acquisition:** Juan David Gutiérrez, Johanna Tapias-Rivera, Martha Liliana Hijuelos-Cárdenas, Ludivia Esther Montaño-Villalba.

**Investigation:** Juan David Gutiérrez, Johanna Tapias-Rivera, Martha Liliana Hijuelos-Cárdenas, Ludivia Esther Montaño-Villalba.

**Methodology:** Juan David Gutiérrez, Johanna Tapias-Rivera, Martha Liliana Hijuelos-Cárdenas, Ludivia Esther Montaño-Villalba.

**Project administration:** Juan David Gutiérrez, Johanna Tapias-Rivera, Martha Liliana Hijuelos-Cárdenas, Ludivia Esther Montaño-Villalba.

**Resources:** Juan David Gutiérrez, Johanna Tapias-Rivera, Martha Liliana Hijuelos-Cárdenas, Ludivia Esther Montaño-Villalba.

**Software:** Juan David Gutiérrez, Johanna Tapias-Rivera, Martha Liliana Hijuelos-Cárdenas, Ludivia Esther Montaño-Villalba.

**Supervision:** Juan David Gutiérrez, Johanna Tapias-Rivera, Martha Liliana Hijuelos-Cárdenas, Ludivia Esther Montaño-Villalba.

**Validation:** Juan David Gutiérrez, Johanna Tapias-Rivera, Martha Liliana Hijuelos-Cárdenas, Ludivia Esther Montaño-Villalba.

**Visualization:** Juan David Gutiérrez, Johanna Tapias-Rivera, Martha Liliana Hijuelos-Cárdenas, Ludivia Esther Montaño-Villalba.

**Writing – original draft:** Juan David Gutiérrez, Johanna Tapias-Rivera, Martha Liliana Hijuelos-Cárdenas, Ludivia Esther Montaño-Villalba.

**Writing – review & editing:** Juan David Gutiérrez, Johanna Tapias-Rivera, Martha Liliana Hijuelos-Cárdenas, Ludivia Esther Montaño-Villalba.

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
