## [Decision Letter · Decision Letter 0]

16 Dec 2025

PGPH-D-25-03098

Effect of sea surface temperature in El Niño regions on dengue dynamics in Colombia: Evidence from causal machine learning

Dear Dr. Juan David Gutierrez

Thank you for submitting your manuscript to PLOS Global Public Health. After careful consideration, we feel that it has merit but does not fully meet PLOS Global Public Health’s publication criteria as it currently stands. Therefore, we invite you to submit a revised version of the manuscript that addresses the points raised during the review process.

We look forward to receiving your revised manuscript.

Kind regards,

Srinivasa Rao Mutheneni, PhD

Academic Editor

Journal Requirements:

1. Please update your online Competing Interests statement. If you have no competing interests to declare, please state: “The authors have declared that no competing interests exist.”

2. Please provide separate main figure files in .tif or .eps format only and remove any figures embedded in your manuscript file. Please also ensure that all files are under our size limit of 10MB. Please leave the figure captions in the manuscript.

3. Please ensure that you refer to Figure 5 in your text as, if accepted, production will need this reference to link the reader to the figure.

4. Some material included in your submission may be copyrighted. According to PLOS’s copyright policy, authors who use figures or other material (e.g., graphics, clipart, maps) from another author or copyright holder must demonstrate or obtain permission to publish this material under the Creative Commons Attribution 4.0 International (CC BY 4.0) License used by PLOS journals. Please closely review the details of PLOS’s copyright requirements here: PLOS Licenses and Copyright. If you need to request permissions from a copyright holder, you may use PLOS's Copyright Content Permission form.

Potential Copyright Issues:

Figure 1: Please confirm whether you drew the images / clip-art within the figure panels by hand. If you did not draw the images, please provide (a) a link to the source of the images or icons and their license / terms of use; or (b) written permission from the copyright holder to publish the images or icons under our CC-BY 4.0 license. Alternatively, you may replace the images with open source alternatives. See these open source resources you may use to replace images / clip-art:

- https://openclipart.org/

Additional Editor Comments (if provided):

Reviewers' comments:

Reviewer's Responses to Questions

**Comments to the Author**

1. Does this manuscript meet PLOS Global Public Health’s publication criteria? Is the manuscript technically sound, and do the data support the conclusions? The manuscript must describe methodologically and ethically rigorous research with conclusions that are appropriately drawn based on the data presented.

Reviewer #1: Yes

2. Has the statistical analysis been performed appropriately and rigorously?

Reviewer #1: Yes

3. Have the authors made all data underlying the findings in their manuscript fully available (please refer to the Data Availability Statement at the start of the manuscript PDF file)?

Reviewer #1: Yes

4. Is the manuscript presented in an intelligible fashion and written in standard English?

Reviewer #1: Yes

5. Review Comments to the Author

Reviewer #1: Dear Authors,

Thank you for submitting your manuscript “Effect of sea surface temperature in El Niño regions on dengue dynamics in Colombia: Evidence from causal machine learning” to PLOS Global Public Health.

Overall, this is a well-structured and rigorous study addressing an important topic at the intersection of climate science, epidemiology, and causal machine learning. The manuscript is clearly written, the methodological framework is strong, and the results are well presented. Your use of DML, DAG-based adjustment, and robustness tests demonstrates methodological sophistication.

My final recommendation is Accept after Minor Revision. Below, I detail both your already-addressed comments and several additional minor improvements that would strengthen clarity, transparency, and reproducibility.

General Assessment

Strengths

The study uses a large temporal and spatial dataset (1,044 municipalities, 2013–2023).

The causal machine learning approach is innovative relative to most ENSO–dengue literature.

The robustness tests offer valuable insights about remaining bias.

The manuscript is detailed and transparent, especially in the Methods section.

Figures are clear and generally well designed.

Areas for Minor Improvement

While the paper is strong, a few minor issues require clarification or polishing to increase readability and rigor. These are not major scientific concerns, but they will improve the manuscript's overall quality.

Specific Minor Revisions

1. Improve the flow and clarity of the Introduction

The Introduction is long and very method-heavy. Consider reorganizing:

Start with the dengue–climate relationship.

Then discuss SST and ENSO relevance.

Then introduce the novelty of causal ML approaches.

Some references appear duplicated (e.g., ENSO–dengue relationships). Confirm consistency.

2. Provide clearer justification for excluding altitude > 2,200 m

You mention the threshold, but the Methods would benefit from a short sentence referencing biological limits of Aedes aegypti, or citing a specific study validating this elevation threshold.

3. Improve Table 1 description

Add a sentence explaining the role of each index more clearly (e.g., why North Atlantic SST is relevant as a confounder). This helps non-climate-experts.

4. Clarify the collider reasoning

You identified MPI, rainfall, temperature, etc. as colliders and excluded them to avoid collider bias.

This is excellent but unusual for most readers; please:

briefly explain the intuition (e.g., “conditioning on variables influenced jointly by exposure and confounders induces bias”).

cite a standard causal inference reference.

5. Expand on the potential mechanism behind altitude-related CATE

You correctly state that the pattern is “not clear”. I recommend adding:

either possible hypothesis (vector biology, indoor–outdoor behavior, socio-economic gradients), or stating more explicitly that this should be part of future work.

6. Improve the explanation of the negative control exposure

The description is technically correct, but consider adding:

one sentence describing why “future SST cannot affect past dengue cases.”

7. Figures – enhance labeling

Figure 4: Add exact SD values in the caption for each El Niño region.

Figure 5: Indicate the altitude range in the figure (e.g., min–max).

8. Clarify limitations related to passive surveillance

The limitations section mentions SIVIGILA constraints, but it would help to:

emphasize variability in laboratory confirmation rates between years.

add one sentence about underreporting during healthcare disruptions.

9. Discussion – reduce repetition

The discussion repeats the ENSO–dengue relationship several times. Consider merging overlapping paragraphs for conciseness.

10. Typographical and minor language edits

Examples:

Line 169: “avoiding the induction of other forms of bias” → “to avoid inducing other forms of bias.”

Line 422: “not clear what the mechanism…” → “the underlying mechanism remains unclear.”

Check spacing inconsistencies around parentheses and units.

Optional but Helpful Additions

These are optional, but could improve the manuscript:

Add a short data availability note within the Results section referencing the GitHub repository.

Include a map of Colombia showing municipalities included/excluded after altitude filtering.

Add citations related to SST impacts on ocean–climate teleconnections in Latin America.

Add this recent study to your references: Shabani, F., Raie, M., & Kabiri, K. (2025). Studying the Impact of Sea Surface Temperature (SST) Changes and El Niño Phenomenon on Coral Reefs Bleaching around Kish Island in the Northern Persian Gulf: A Remote Sensing Approach. Deep Sea Research Part II: Topical Studies in Oceanography, 105550.

Overall

Your paper is scientifically valuable and nearly ready for publication. I am confident that addressing these minor points will strengthen the manuscript further.

Recommendation: Accept after Minor Revision.

6. PLOS authors have the option to publish the peer review history of their article (what does this mean?). If published, this will include your full peer review and any attached files.

**Do you want your identity to be public for this peer review?** For information about this choice, including consent withdrawal, please see our Privacy Policy.

Reviewer #1: **Yes:**Keivan Kabiri

Figure Resubmissions: While revising your submission, we strongly recommend that you use PLOS’s NAAS tool (https://ngplosjournals.pagemajik.ai/artanalysis) to test your figure files. NAAS can convert your figure files to the TIFF file type and meet basic requirements (such as print size, resolution), or provide you with a report on issues that do not meet our requirements and that NAAS cannot fix.

---

## [Decision Letter · Decision Letter 1]

6 Jan 2026

Effect of sea surface temperature in El Niño regions on dengue dynamics in Colombia: Evidence from causal machine learning

PGPH-D-25-03098R1

Dear Juan David Gutierrez,

We are pleased to inform you that your manuscript 'Effect of sea surface temperature in El Niño regions on dengue dynamics in Colombia: Evidence from causal machine learning' has been provisionally accepted for publication in PLOS Global Public Health.

Best regards,

Srinivasa Rao Mutheneni, PhD

Academic Editor

Reviewer Comments (if any, and for reference):

Reviewer's Responses to Questions

**Comments to the Author**

1. If the authors have adequately addressed your comments raised in a previous round of review and you feel that this manuscript is now acceptable for publication, you may indicate that here to bypass the “Comments to the Author” section, enter your conflict of interest statement in the “Confidential to Editor” section, and submit your "Accept" recommendation.

Reviewer #1: All comments have been addressed

2. Does this manuscript meet PLOS Global Public Health’s publication criteria? Is the manuscript technically sound, and do the data support the conclusions? The manuscript must describe methodologically and ethically rigorous research with conclusions that are appropriately drawn based on the data presented.

Reviewer #1: Yes

3. Has the statistical analysis been performed appropriately and rigorously?

Reviewer #1: Yes

4. Have the authors made all data underlying the findings in their manuscript fully available (please refer to the Data Availability Statement at the start of the manuscript PDF file)?

Reviewer #1: Yes

5. Is the manuscript presented in an intelligible fashion and written in standard English?

Reviewer #1: Yes

6. Review Comments to the Author

Reviewer #1: The current version of MS is acceptable fpr publication on PGPH

7. PLOS authors have the option to publish the peer review history of their article (what does this mean?). If published, this will include your full peer review and any attached files.

**Do you want your identity to be public for this peer review?** For information about this choice, including consent withdrawal, please see our Privacy Policy.

Reviewer #1: **Yes:**Keivan Kabiri
